# Metformin Inhibited GSDME to Suppress M2 Macrophage Pyroptosis and Maintain M2 Phenotype to Mitigate Cisplatin-Induced Intestinal Inflammation

**DOI:** 10.3390/biomedicines12112526

**Published:** 2024-11-04

**Authors:** Ke Jiang, Qi He, Chenhui Wang, Wen Yang, Changdong Zhou, Jian Li, Jiangbo Li, Yuke Cui, Jingqi Shi, Zhenqiao Wei, Yuanyuan Jiao, Ligai Bai, Shengqi Wang, Liang Guo

**Affiliations:** 1College of Pharmaceutical Sciences, Hebei University, Baoding 071002, China; 2Bioinformatics Center of AMMS, Beijing 100850, China

**Keywords:** cisplatin, gasdermin E, M2 macrophages, pyroptosis, metformin, intestinal inflammation

## Abstract

Background: The continuous clinical use of cisplatin is prevented by gastrointestinal toxicity. Methods: Cisplatin was used to treat THP-1-derived macrophages to see its differential effects on different subtypes of macrophages. Wild-type and Gsdme^−/−^ mice models were used to examine the effect of cisplatin and metformin on intestinal inflammation in vivo. The effect of GSDME on macrophage polarization was further confirmed by GSDME knockdown. Results: We found that M2 macrophages, with more cell blebbing and GSDME cleavage, were more sensitive to cisplatin-induced pyroptosis than M1 macrophages. Cisplatin was capable of enhancing the M1 phenotype, which was reversed by GSDME knockdown. GSDME contributed to M1 polarization and GSDME knockdown promoted M2 phenotype via STAT6 activation. Reduced intestinal inflammation and increased M2 macrophage numbers was detected in cisplatin-treated GSDME-knockout mice. Furthermore, metformin alleviated cisplatin-induced intestinal inflammation by reducing M2 pyroptosis and enhancing M2 phenotype through GSDME inhibition. Conclusion: This is the first study to reveal the non-pyroptotic role of GSDME in macrophage polarization, revealing that metformin could be used in combination with cisplatin to reduce intestinal toxicity.

## 1. Introduction

Cisplatin is widely used in treating multiple types of tumors because of its potency in inducing cell death. However, the severe damage to the intestinal mucosa induced by cisplatin usually prevents chemotherapy continuation and causes bacteremia, which can be life-threatening [1]. Thus, it is of great significance to understand the molecular mechanism of cisplatin-induced intestinal injury.

The homeostasis within the intestine is largely maintained by macrophages, whose dysregulation presents as the central feature of the inflammatory bowel disease [2]. Macrophages are key innate immune cells with high plasticity to transdifferentiate between classically activated (M1) and alternatively activated (M2) types to regulate tissue repair and regeneration in response to various stimuli [3,4,5]. Pro-inflammatory M1 macrophages contribute to the progression of acute intestinal injury, while M2 macrophages, considered to exhibit anti-inflammation roles, mainly function in tissue repair to reduce intestinal inflammation [4,6,7,8]. Therefore, the modulation of the balance of M1/M2 subtypes is critical for intestinal homeostasis.

Cisplatin-induced therapeutic and adverse effects have been revealed to be closely associated with macrophage participation [9,10,11,12,13,14], with the underlying molecular mechanisms largely unknown. The uptake of cisplatin in macrophage subtypes at the single-cell level was recently examined by a high-throughput laser ablation–inductively coupled plasma–time-of-flight mass spectrometry (LA-ICP-TOFMS) method to show a higher incorporation of cisplatin in the M2 subtype than in the M1 phenotype [15], which indicated the differential effects of cisplatin on different subtypes of macrophages. Cisplatin is capable of inducing cell pyroptosis by cleaving gasdermin proteins such as gasdermin D (GSDMD) and gasdermin E (GSDME) to induce plasma membrane blebbing and pore-formation [16,17,18]. The role of gasdermin proteins in pyroptosis has been well-established. However, more recently, the non-pyroptotic function of GSDMD is also reported. GSDMD phosphorylates eIF2α and activates the endoplasmic reticulum (ER) stress response to enhance cisplatin-induced apoptosis [19]. Moreover, differential cleavage of GSDMD acts as a hub regulating immunity versus tolerance in the small intestine by inducing the transcription of Class II Major Histocompatibility Complex Transactivator (CIITA) and Major Histocompatibility Complex Class II (MHCII) molecules [19]. The pyroptotic cells are characterized by the release of a large number of inflammatory factors, including interleukin-1ß (IL-1ß), while macrophage polarization, especially M1 polarization, is also accompanied by proinflammatory cytokine secretion, which implies a potential association between pyroptosis and macrophage transdifferentiation or that common regulators might function in both processes. However, no related data have been reported.

Metformin has been revealed to enhance the therapeutic effect of cisplatin. For example, metformin increases the anti-cancer effects of cisplatin in meningioma, augments the radiosensitization and chemosensitization of non-small cell lung cancer (NSCLC) cells to cisplatin, and prevents resistance to cisplatin in NSCLC and triple negative breast cancer cells [20,21,22,23,24]. Besides the synergistical effect, metformin is also endowed with anti-inflammation roles, with an emphasis on intestinal inflammatory injury [25]. However, no data have been reported on the potential link between cisplatin-induced intestinal damage and metformin.

In this study, we found that M2 macrophages were more sensitive to cisplatin treatment, displaying a more pyroptotic phenotype change than M1 macrophages, which could be reversed by metformin through inhibition of the caspase-3/GSDME pathway. More importantly, GSDME contributed to M1 polarization and knockdown of GSDME promoted the M2 phenotype via STAT6 activation, which was further evidenced by an increasing number of M2 macrophages, as well as reduced intestinal inflammation, in cisplatin-treated GSDME-knockout mice. Metformin alleviated cisplatin-induced intestinal damage by inhibition of GSDME, which simultaneously reduced M2 macrophage death and enhanced the M2 phenotype. Our study sheds new light on the molecular mechanism of the metformin-mitigated intestinal damage induced by cisplatin, revealing, for the first time, the potential association between pyroptosis and macrophage polarization by GSDME.

## 2. Materials and Methods

### 2.1. Cell Culture and Treatment

The human macrophage THP-1 cells (ATCC) were cultured in RPMI-1640 medium supplemented with 10% FBS, 100 g/mL streptomycin, and 100 U/mL penicillin. All cells were grown at 37 °C with 5% CO_2_.

THP-1 monocytes were differentiated into a resting state macrophage phenotype, referred to as M0, by incubating the cells in 20 ng/mL PMA (Beyotime, Cat# S1819, Shanghai, China) at a density of 5 × 10^5^ cells/mL for 24 h. Differentiated cells were washed with culture medium and rested for another 24 h in the culture medium to obtain the resting state of macrophages (M0). The resting macrophages (M0) were cultured for 48 h with fresh medium supplemented with 20 ng/mL IFN-γ (SinoBiological, Cat# 11725-HNAE, Beijing, China) and 100 ng/mL LPS (Thermo Fisher Scientific, Cat# 00-4976-93, Waltham, MA, USA) to differentiate them into the M1 phenotype or with 20 ng/mL IL-4 (SinoBiological, Cat# 11846-HNAE) and IL-13 (SinoBiological, Cat# 10369-HNAC) to differentiate them into the M2 phenotype.

M0, M1, and M2-like macrophages were cultured in serum-free medium after washing twice with serum-free medium. Metformin (MedChemExpress, Cat# HY-17471A, Princeton, NJ, USA) or Z-VAD-FMK (Beyotime, Cat# C1202) was added to the serum-free medium at a specified concentration. Cisplatin (MedChemExpress, Cat# HY-17394) was added to the cultures after 1 h at a concentration of 15 μmol/L; following a 24 h incubation period, the supernatant was collected and the cells harvested

### 2.2. Mouse Models

Male C57BL/6 mice (6 week of age and 18–20 g body weight) were purchased from SiPeiFu Laboratories (License No. SCXK- (jing) 2019-0010).

For metformin treatment, mice were injected intraperitoneally with metformin (at doses of 100 and 200 mg/kg) 3 days prior to receiving cisplatin, and mice were sacrificed 3 days later.

Gsdme^−/−^ mice were maintained in our laboratory. Gsdme^−/−^ mice were generated by co-microinjection of in vitro-translated Cas9 mRNA and gRNAs into C57BL/6 zygotes, as described for Gsdmd^−/−^ mice [26]. All animal experiments complied with the Ministry of Health national guidelines concerning the housing and care of laboratory animals and were performed in alignment with institutional policies. The experimental protocols were meticulously reviewed and subsequently approved by the Institutional Animal Care and Use Committee at the National Institute of Biological Sciences prior to commencement of the studies.

### 2.3. Cell Viability

Cell viability was evaluated using the MTT kit (Beyotime) according to the manufacturer’s instructions. Briefly, after macrophage confluence reached about 80%, the cells were pretreated with metformin for 1 h, followed by exposure to cisplatin for 24 h at 37 °C in an atmosphere containing 5% CO_2_. After incubation, the medium was discarded, and 100 μL of fresh medium containing 10% MTT solution was added to each well and then incubated for 4 h at 37 °C. To each well was added 100 μL of formazan solvent before incubation for 3 h at 37 °C. The absorbance (λ = 570 nm) was measured using a microplate reader.

### 2.4. LDH Release Assay

Various types of macrophages were cultured until reaching approximately 80% confluence in 96-well plates. Subsequently, the cells were treated with cisplatin alongside different compound samples (metformin or the pan-caspase inhibitor Z-VAD-FMK). The LDH release of each sample was measured three times using a CytoTox 96^®^Nonradioactive Cytotoxicity Assay kit at various time intervals. A lysis solution was utilized to produce the maximum LDH release (100% LDH). The assay and control wells were allocated across clear 96-well plates, after which 50 μL CytoTox 96^®^ Reagent was added to each well. The wells were shielded from light and left to incubate at room temperature for 30 min. The plates were then analyzed using a BioTek Epoch microplate reader set to a wavelength of 492 nm. The percentage of LDH release was determined using the following formula, LDH release (%) = (Experimental LDH release-Spontaneous LDH release)/(Total LDH release-Spontaneous LDH release) × 100%.

### 2.5. Quantitative Real-Time PCR and Flow Cytometry

RNA was isolated using the RNA Easy Fast Tissue/cell Kit (TIANGEN, Cat# DP451, Beijing, China) according to the manufacturer’s instructions. The RNA concentration was quantified using the Qubit RNA BR Kit (Invitrogen, Cat# Q10210, Waltham, MA, USA). Subsequently, reverse transcription was carried out with a Fasting gDNA Dispelling RT SuperMix Fastking kit (TIANGEN, Cat# KR118) according to the manufacturer’s instructions. RT-PCR was performed using a Talent qPCR PreMix (SYBR Green) kit (TIANGEN, Cat# FP209) according to the manufacturer’s instructions (3 min at 95 °C, followed by 40 cycles of 5 s at 95 °C and 15 s at 60 °C). Target gene expression was normalized to GAPDH gene expression, and each sample was tested in triplicate. Primer sequences are provided in the Appendix A. Primers for genes were specific based on verification from primer-blast, and the melting curve was a single sharp peak.

M2 macrophages were digested using no-EDTA trypsin, then the cells were washed twice with PBS and resuspended in 500 μL of binding buffer. Next, the cells were stained with 5 μL Annexin V-FITC (Beyotime, Cat# C1062S-1) staining solution and 5 μL PI (Beyotime, Cat# C1062S-3) solution in the dark for 15 min at room temperature. The extent of cell pyroptosis was quantitatively assessed through flow cytometric analysis. To detect cell surface markers, cells were stained with CD206 (BioLegend, Cat# 321105, San Diego, CA, USA) antibodies or CD86 (BioLegend, Cat# 374202) antibodies at 4 °C for 30 min. Cellular analyses were performed utilizing a BD FACSAria III Flow Cytometer. Data were analyzed using FlowJo 10 software.

### 2.6. Western Blot Analysis

Cells or mouse intestinal tissues were lysed using RIPA buffer with protease and phosphatase inhibitors cocktails (Thermo Fisher Scientific, Cat# 87786). The protein content was determined using Pierce BCA Protein Assay Reagent (Thermo Fisher Scientific, Cat# 23227) according to the manufacturer’s instructions. Then, 20 μg of total protein from each group was run on SDS-PAGE gels and then transferred to PVDF membranes. The following antibodies were used for hybridization: anti-GSDME (Abcam, Cat# ab215191, Cambridge, UK, 1:1000), anti-cleaved caspase-3 (Beyotime, Cat# AC033, 1:1000), anti-cleaved PARP (Beyotime, Cat# AF1567, 1:2000), and anti-phospho-STAT6 (Beyotime, Cat# AF5950, 1:1000). β-Actin was used as a loading control. The membranes were then washed with TBST and incubated with horse radish peroxidase-conjugated secondary antibodies. The bands were then visualized using a VLBER FUSION FX7 Spectra.

### 2.7. Cell Transfection

The siRNA targeting the GSDME gene, which was produced by GenePharma, was utilized for the purpose of knocking down GSDME expression. When cell confluence reached about 70–80%, the cells were transfected with 10 μM siRNA using Lipofectamine RNAi MAX reagent (Thermo Fisher Scientific) according to the manufacturer’s protocol. The knockdown efficiency of si-GSDME was assessed using qRT-PCR and Western blot analysis. The sense sequences for the siRNA used are presented in Appendix A.

### 2.8. Histology and Immunofluorescence

Intestinal tissues were isolated, fixed with 4% paraformaldehyde, and embedded in paraffin. Sections of 5 μm were cut for hematoxylin and eosin staining following deparaffinization and rehydration. For immunostaining, the sections were put in 3% H_2_O_2_ for 20 min followed by deparaffinization and heat-mediated antigen retrieval treatment and incubated in blocking buffer (3% BSA (Solarbio, Cat# A8020, Beijing, China) in PBS supplemented with 0.1% Triton X-100) at room temperature for 1 h. Sections were incubated with primary antibody for 2 h, followed by detection using HRP-conjugated secondary antibody and TSA-fluorophores. The primary and secondary antibodies were eliminated by heating the slides. F4/80 (Cell Signaling Technology, Cat# 70076S, Boston, MA, USA, 1:1000), CD86 (Cell Signaling Technology, Cat# 19589T, 1:1000), CD206 (Abcam, Cat# ab300621, 1:500), Ly6g (Abcam, Cat# ab25377, 1:500), and GSDME (Abmart, Cat# TD9705, Shanghai, China, 1:1000) were sequentially detected. Vectashield containing DAPI nuclear counterstain was used to mount the sections. The slices were imaged using the confocal laser scanning microscopy platform Zeiss LSM880.

### 2.9. Statistical Analysis

Most data are expressed as the mean ± SD, and statistical analysis of the differences between two groups was performed using two-tailed Student’s t-tests. Differences between multiple groups were analyzed using one-way or two-way ANOVA, followed by Tukey’s or Sidak’s multiple-comparison tests. Statistical analysis was performed using GraphPad Prism 7. * *p* < 0.05 indicated a statistically significant difference, ** *p* < 0.01 indicated a highly significant difference, and *** *p* < 0.001 indicated an extremely significant difference.

## 3. Results

### 3.1. M2 Macrophages Were More Sensitive to Cisplatin-Induced Pyroptotic Change via Activation of caspase3/GSDME Pathway

To compare the effects of cisplatin on the cell death of different macrophage subtypes, M1 and M2 macrophages were polarized from THP-1 cells as previously reported. Briefly, THP-1 cells were stimulated with Phorbol-12-myristate-13-acetate (PMA) for 24 h to become M0 macrophages, followed by treatment with Interferon-gamma (IFN-γ)/Lipopolysaccharide (LPS) or interleukin (IL)-4/IL-13 to polarize M0 macrophages into the M1 or M2 phenotype, respectively (Appendix A). M1 and M2 macrophages exhibited differential morphological changes, with M1 macrophages displaying an elongated spindle-like shape and M2 macrophages being rounded (Appendix A). Interleukin-6 (IL-6) and tumor necrosis factor alpha (TNF-α), the characterized cytokines for proinflammatory macrophages, were significantly induced in the M1 subtype, whereas Arg-1, CD163, and CD206 were increased in M2 macrophages (Appendix A). These results collectively indicated the successful polarization of THP-1 cells into M1- and M2-polarized macrophages.

Then, we evaluated the cytotoxicity effect of cisplatin on M1 and M2 macrophages via MTT assay. After 24 h of treatment, the IC50 value of cisplatin was about 18.72 µM in M1 macrophages and 20.7 µM in M2 macrophages, indicating that cisplatin induced comparable cell death rates in both phenotypes (Appendix A). Intriguingly, the pyroptotic phenotype change was extremely obvious in M2 macrophages. The typical characteristics of pyroptosis, such as cell blebbing and lactate dehydrogenase (LDH) secretion [27], were observed in M2 macrophages in a dose-dependent manner in response to cisplatin, with a concomitantly higher percentage of an Annexin V^+^/PI^+^ cell population (Figure 1A–D).

Cisplatin has been reported to induce pyroptosis via the caspase-3/GSDME signaling pathway [17]. Western blot analysis revealed that cisplatin induced a higher level of cleaved caspase-3 and GSDME in M2 macrophages compared to that in the M0 and M1 subtypes (Figure 1E). Of note, cleaved PPAR, a key regulator of apoptosis, was also dramatically induced in M2 macrophages, which indicated that cisplatin induced more than one type of cell death in M2 macrophages. Of note, cisplatin was also capable of inducing caspase-3 and GSDME cleavage in M1 macrophages, though at a much lower level than in M2 macrophages, implying the differential effects of cisplatin in different macrophage subtypes and that the M2 phenotype was more sensitive to cisplatin-induced pyroptosis.

To better illustrate the regulatory effect of caspase-3 on the cisplatin-induced cell death of macrophages, different subtypes of macrophages were pretreated with a pan-caspase inhibitor (Z-VAD-FMK) for one hour, followed by administration of cisplatin for 24 h. Cell blebbing in M2 macrophages was inhibited by Z-VAD-FMK, with a significant reduction in the percentage of the Annexin V^+^/PI^+^ population upon cisplatin treatment. (Figure 1F). Additionally, cisplatin-activated LDH release and GSDME cleavage were also dramatically suppressed by Z-VAD-FMK in M2 macrophages compared to the suppression in M0 and M1 macrophages (Figure 1G,H).

Our data suggest that M1 and M2 macrophages exhibited differential sensitivities to cisplatin-induced cell-death and that M2 macrophages were more susceptible to cisplatin-induced pyroptosis through the caspase-3/GSDME pathway.

### 3.2. GSDME Played Potential Roles in Macrophage Polarization

The non-pyroptotic functions of the gasdermin protein GSDMD have only just begun to be unveiled [28], while no such effect for GSDME has been reported. After peritoneal injection of cisplatin into GSDME-knockout mice and wild-type mice (Figure 2A), we found that cisplatin-induced colon length reduction was significantly reversed in GSDME-knockout mice (Figure 2B). Moreover, consistent with a previous report [29], GSDME-knockout mice exhibited a much lower level of intestinal inflammation upon cisplatin treatment, as evidenced by reduced IL-6 and TNF-α expression and lower infiltration of leukocytes (Figure 2C,D). GSDME-knockout mice were more resistant to cisplatin-induced intestinal injury, as the surviving crypts and villi were not significantly affected by cisplatin (Figure 2E,F). Moreover, fewer CD86^+^/F4/80^+^ M1 macrophages were detected in the small intestine tissue of GSDME-knockout mice compared to the wild-type mice upon cisplatin treatment, with the number of CD206^+^/F4/80^+^ M2 macrophages dramatically increased, which indicated GSDME-deficiency contributed to the M2 polarization of the macrophages in the small intestine (Figure 2G,H).

These data indicate that there might be potential links between GSDME and macrophage polarization. Thus, we evaluated the potential role of GSDME in regulating the phenotype of macrophage subtypes. For M1 macrophages, the expression of the marker genes IL-6 and TNF-α were enhanced by cisplatin, while they were significantly reversed by GSDME knockdown. Conversely, the expression of M2 markers (Arg-1, CD163) in the M1 macrophages was further reduced by cisplatin, while mildly increased by GSDME knockdown. The percentage of CD86^+^ M1 macrophages that was increased by cisplatin was reduced by GSDME siRNA (Figure 3A,B). For M2 macrophages, cisplatin-induced reduction of the expression of the M2 marker genes CD163 and Arg and the CD206^+^ population percentage were increased by GSDME knockdown, whereas cisplatin-increased IL-6 and TNF-α expression was reduced. Additionally, cisplatin directly induced the expression of IL-6 and TNF-α and in M0 macrophages, which could also be reduced by GSDME knockdown (Figure 3C,D). Moreover, GSDME inhibition also mildly restored cisplatin-induced repression of M2 signature genes in M0 macrophages (Appendix A).

Collectively, we found that cisplatin was potent in inducing the M1 polarization of macrophages, during which GSDME played critical roles in maintaining the proinflammatory phenotype of M1 macrophages. This study is the first time that GSDME has been revealed to function in a cellular process other than pyroptosis, indicating a potential link between GSDME-mediated pyroptotic change and macrophage polarization.

### 3.3. Metformin Suppressed GSDME Cleavage to Inhibit M2 Pyroptosis and Reverse the M2 Phenotype in Vitro

Metformin has been reported to play synergistical roles with cisplatin in treating multiple types of cancers, as well as to exhibit anti-inflammation effects. However, it has not been studied if metformin is efficacious to combat cisplatin-induced intestinal damage. Since M2 macrophages were more sensitive to cisplatin-induced cell death, firstly, we evaluated the effect of metformin on the M2 cell death caused by cisplatin. Metformin significantly reduced cisplatin-induced cell blebbing (Figure 4A) and LDH secretion (Figure 4B). In addition, the percentage of the pyroptotic cell population (Annexin V^+^/PI^+^) was also downregulated and pretreatment with metformin decreased the percentage of double-positive cells (Figure 4C). Western blot analysis showed that metformin markedly reduced cisplatin-induced GSDME cleavage in the M2 macrophages (Figure 4D), which indicated that metformin was capable of protecting M2 macrophages from cisplatin-induced cell death.

Furthermore, in M2 macrophages upon metformin treatment, cisplatin-increased TNF-α and IL-6 were downregulated by metformin, while cisplatin-induced reduction of CD163 and Arg-1 were restored (Figure 4E). STAT6 phosphorylation has been proven to be capable of regulating M2 polarization [30]. We found that the expression of phosphorylated-STAT6 (p-STAT6) was reduced by cisplatin to weaken the M2 phenotype, which could be reversed by both metformin and GSDME siRNA. Of note, both metformin and GSDME siRNA reduced GSDME cleavage, as well as increased the phosphorylation of STAT6, to comparable levels, which indicated that the pro-M2 role of metformin was probably mediated by GSDME cleavage (Figure 4F).

Collectively, our data revealed that metformin was potent in mitigating cisplatin-induced M2 pyroptosis and M1 polarization through the inhibition of GSDME cleavage.

### 3.4. Metformin Interrupted Cisplatin-Induced M1 Polarization via GSDME in Vivo

Subsequently, we analyzed the effect of metformin on macrophage polarization in mice models upon cisplatin treatment. Amounts of 100 or 200 mg/kg metformin were intraperitoneally injected into the mice before cisplatin treatment (Figure 5A). Cisplatin-induced colon length reduction was significantly reversed by metformin (Figure 5B). Metformin was capable of downregulating the cleavage of GSDME and the increased level of IL-6 and TNF-α in the intestine tissue induced by cisplatin (Figure 5C,D). The percentage of F4/80+/CD86+ M1 macrophages in the intestine tissue was dramatically increased by cisplatin and reversed by metformin. Correspondingly, metformin increased the percentage of F4/80^+^/CD206^+^ M2 macrophages upon cisplatin treatment (Figure 5E–I). The intestinal villi and crypts are sensitive to intestinal injuries [31]. Compared to the control group, the integrity and numbers of the villi and crypts of the intestine derived from the cisplatin-treated mice were detected to be altered (with an obvious infiltration of Ly6G^+^ neutrophils), and this could be significantly reversed by metformin (Appendix A). Of note, upon cisplatin treatment, no significant difference in the colon length, the numbers of surviving villi and crypts, or the amount inflammatory cytokines (IL-6, TNF-α) was found between GSDME^−/−^ mice treated with metformin or not, which indicated again that the metformin-alleviated intestine inflammatory injury was mediated by GSDME (Appendix A).

Our results implied that metformin was potent in mitigating cisplatin-induced intestinal inflammatory damage via suppressing the pyroptosis of M2 macrophages and enhancing M2 polarization through GSDME.

## 4. Discussion

Pyroptosis is a highly inflammatory form of programmed necrosis [32] and has been regarded as one of the main reasons for cisplatin-induced intestinal damage. Macrophages, especially the balance between the M1 and M2 phenotypes, are critical for the maintenance of gastrointestinal homeostasis. Classically activated M1 cells are involved in initiating and maintaining inflammation, while alternately activated M2 cells are related to anti-inflammation processes [33]. Therefore, to some extent, the polarization of M1 macrophages and pyroptosis have multiple similarities in inducing inflammatory reactions. However, no data regarding the potential links between macrophage pyroptosis and polarization have been reported. In this study, we mainly explored the differential effect of cisplatin on different subtypes of macrophages; we found that M2 macrophages were more sensitive to cisplatin-induced pyroptosis than M1 macrophages and that cisplatin was potent in inducing M1 polarization, which was probably the leading cause for cisplatin-induced intestinal damage. We, for the first time, revealed that GSDME was critical for the maintenance of the M1 phenotype and the GSDME knockdown promoted M2 polarization, alleviating cisplatin-induced excessive inflammation in the intestine. Metformin was efficient at mitigating cisplatin-induced intestinal damage by inhibiting GSDME to simultaneously reduce M2 pyroptosis and enhance the M2 phenotype within the intestine tissue (Figure 6). Our study revealed the potential role of GSDME in macrophage polarization and proposed that metformin was able to reduce cisplatin-induced intestinal damage via the inhibition of GSDME.

Cisplatin is capable of inducing cell pyroptosis [11,34,35,36,37], however, no further data have been reported regarding its differential effect on the different types of macrophages. We found that, though cisplatin induced comparable cell death rates in both macrophage phenotypes, M2 macrophages were found to be more sensitive to cisplatin-induced pyroptosis, evidenced by more cell blebbing and GSDME cleavage. Moreover, compared to the wild-type mice, the GSDME-knockout mice, exhibiting more M2 macrophages within the intestine tissue, were found to be more resistant to cisplatin-induced intestinal damage, which indicated GSDME probably exhibited dual effects on both pyroptosis and macrophage polarization.

With the extensive study of gasdermin-mediated pyroptosis, the non-pyroptotic functions of pyroptosis-associated molecules are being unveiled. GSDMD was reported to enhance cisplatin-induced apoptosis by phosphorylating eIF2α and activating the endoplasmic reticulum (ER) stress response [19]. Differential cleavage of GSDMD was reported to regulate the immune balance in the small intestine by inducing the transcription of CIITA and MHCII molecules [38]. The latest research by Zhexu Chi et al. has suggested that in hyperactive macrophages, the GSDMD-guided metabolic crosstalk between macrophages and muscle stem cells was responsible for the tissue repair process [39]. These data indicated that, besides inducing cell death, the pyroptotic phenotypic change or pyroptosis-associated molecules play critical roles in other cellular processes. Besides the established role in pyroptosis [40,41,42,43], no data have been published about the non-pyroptotic role of GSDME. By downregulating GSDME expression via siRNA, we confirmed that GSDME was vital for the maintenance of the M1 phenotype, while knockdown of GSDME enhanced the phenotypic characteristics of M2 macrophages.

The continuous clinical application of cisplatin is significantly restricted by the induction of severe adverse reactions, including the gastrointestinal toxicity; thus, it would be beneficial to use cisplatin in combination with other reagents to reduce its side effects while maintaining or even enhancing its therapeutic effects. Metformin, in both prospective and retrospective studies, was revealed to be associated with anti-neoplastic and anti-inflammation activity in the treatment of various types of tumors [44,45]. Combined regimens of cisplatin and metformin have been suggested to exhibit synergistic or additive effect to increase treatment efficacy in the case of reducing tumor volume and minimizing side effects [22,46]. Yet, no data have been published about the role of metformin in cisplatin-induced gastrointestinal toxicity. In this study, we found that cisplatin was capable of augmenting the M1 phenotype to play pro-inflammatory roles within the intestine tissue, while metformin could dramatically reduce cisplatin-induced severe inflammation in the intestine. M1 and M2 macrophages have been related to controversial roles in the process of inflammation, with M1 macrophages exhibiting proinflammatory and M2 anti-inflammatory roles [47,48]. Metformin efficiently inhibited the pyroptosis of M2 macrophages by suppressing the activation of the caspase-3/GSDME pathway and reduced M1-macrophage activation within the intestines of cisplatin-treated mice, restoring the length and the number of surviving villi and crypts in the intestine. Notably, GSDME knockdown increased the expression of p-STAT6, the critical regulator of M2 polarization [38,49], to a level comparable with that of the metformin-treated cells. Moreover, in GSDME knockout mice, no significant difference was observed between the mice treated with or without metformin. These data indicate that the role of metformin in reducing cisplatin-induced M1 polarization was mediated by its inhibition of GSDME.

Though we have unveiled the potential role of GSDME in macrophage polarization and the beneficial effect of metformin on cisplatin-induced intestinal inflammation through GSDME, there are still some limitations to this study: (1) the molecular mechanism of the role of metformin in inhibiting cisplatin-induced GSDME activation and the role of GSDME, or pyroptotic-like change, in macrophage polarization are worth deep study; (2) reagents targeting pyroptosis or molecules involved in pyroptotic change are probably beneficial in reducing cisplatin-induced intestinal inflammation and might be promising in clinical use, which also needs to be extensively studied; and (3) the used dose of metformin or the ratio of metformin to cisplatin should be clarified to obtain an optimal effect clinically.

In summary, our results revealed that cisplatin exhibited differential effects on different subtypes of macrophages, with M2 macrophages being more sensitive to cisplatin-induced pyroptosis and the M1 phenotype enhanced through GSDME cleavage. Metformin efficiently mitigated cisplatin-induced intestinal injury through reducing M2 pyroptosis and elevating the M2 phenotype within the intestine tissue. This is the first time that GSDME has been revealed to function in macrophage polarization. Our study has shed new light on the link between pyroptotic change and other cellular processes and the potential use of metformin in mitigating cisplatin-induced intestinal damage via GSDME.

## 5. Conclusions

Our study revealed that cisplatin could induce a more obvious pyroptotic phenotype change in M2 macrophages than that in M1 macrophages, which could be reversed by metformin through inhibition of the caspase-3/GSDME pathway. More importantly, GSDME was, for the first time, endowed with a non-pyroptotic role. GSDME contributed to M1 polarization and the knockdown of GSDME promoted the M2 phenotype via STAT6 activation, which was further demonstrated by an increasing number of M2 macrophages, as well as reduced intestinal inflammation, in GSDME-knockout mice upon cisplatin treatment. Metformin mitigated cisplatin-induced intestinal damage by inhibiting GSDME cleavage, which reduced M2 macrophage death and enhanced the M2 phenotype at the same time. Our study sheds new light on the protective effect of metformin against cisplatin-induced intestinal damage, along with the underlying mechanism, to reveal a potential association between pyroptosis and macrophage polarization by GSDME.

## Figures and Tables

**Figure 1 biomedicines-12-02526-f001:**
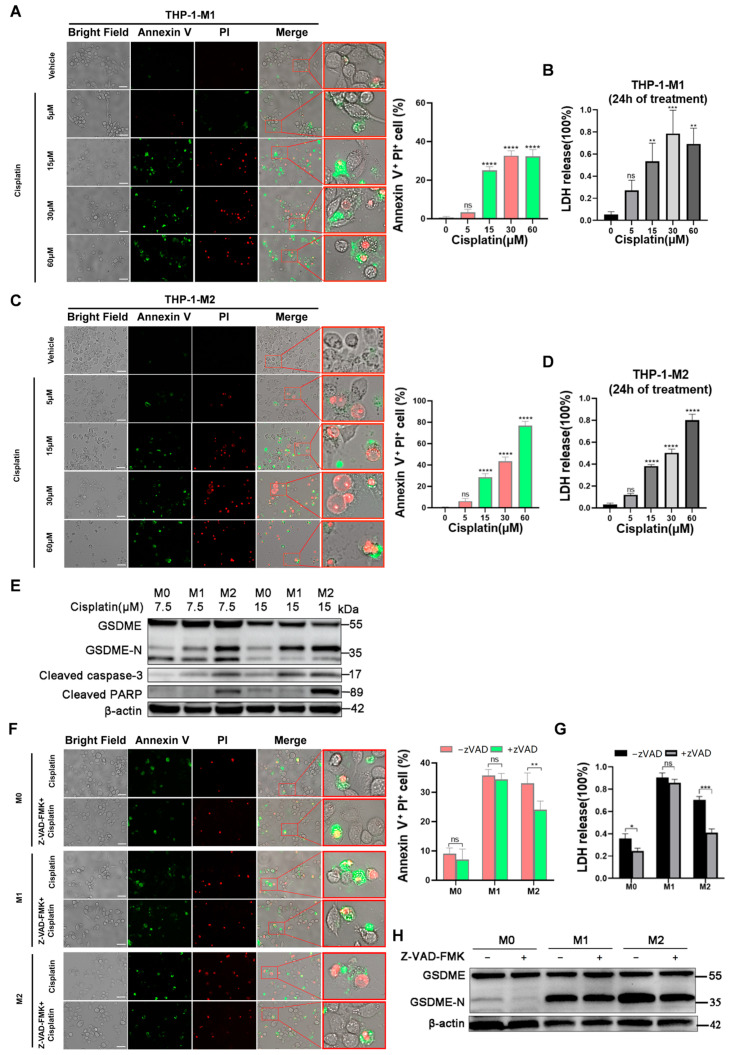
The differential effects of cisplatin on different types of macrophages. (**A**,**C**) Representative images of the bright field and immunofluorescent (IF) assay of THP-1-derived M1 (**A**) or M2 (**C**) macrophages treated with or without cisplatin. THP-1 cells were polarized into M1 or M2 macrophages and treated with or without cisplatin at different concentrations for 24 h, followed by an immunofluorescent assay of Annexin V (green) and Propidium Iodide (PI, red). For the merged images, the right panels show higher-magnification views of the boxed areas in the left panels (40×). The average percentages of Annexin V^+^/PI^+^ positive cells were examined using triplicate experiments. Scale bar, 20 μm. (**B**,**D**) The LDH release of THP-1-derived M1 and M2 macrophages treated with or without cisplatin at different concentrations. The experiments were performed in triplicate. (**E**) Western blot of GSDME, activated caspase 3, and activated PARP expression in macrophages treated with cisplatin. THP-1-derived M1 and M2 macrophages were treated with cisplatin at 7.5 and 15 μM for 24 h, followed by Western blot of the indicated proteins. (**F**–**H**) The effect of Z-VAD-FMK on the cisplatin-induced cell death of different subtypes of macrophages. Different THP-1-derived subtypes of macrophages were pretreated with the pan-caspase inhibitor Z-VAD-FMK for one hour, followed by administration of cisplatin for 24 h. (**F**) The representative images of the immunofluorescent assay of macrophages and the average percentage of Annexin V^+^/PI^+^ positive examined by triplicate experiments. Scale bar, 20 μm. (**G**) The LDH release was analyzed using triplicate experiments. (**H**) The Western blot analysis of GSDME cleavage of different subtypes of macrophages. The data are shown as mean ± SD. *p* values were calculated via one-way ANOVA or two-way ANOVA with Dunnett’s test. * *p* < 0.05, ** *p* < 0.01, *** *p* < 0.001, **** *p* < 0.0001, and ns (not significant).

**Figure 2 biomedicines-12-02526-f002:**
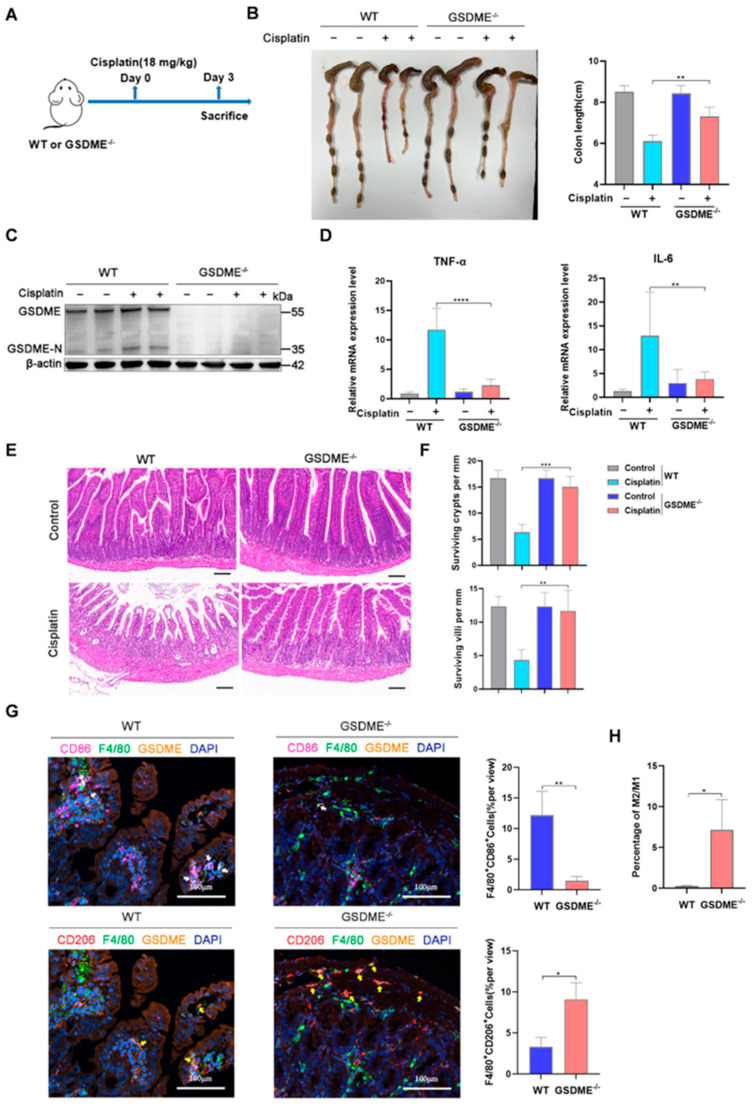
GSDME-knockout mice were resistant to cisplatin-induced intestinal damage. (**A**) Schematics of cisplatin treatment of the wild-type (WT) and GSDME-knockout (GSDME^−/−^) mice. The WT and GSDME^−/−^ mice were intraperitoneally injected with cisplatin for 3 days, followed by subsequent assays. (**B**) The representative images and statistical analysis of the colon length from different groups (*n* = 6). (**C**) The Western blot of GSDME expression among different groups. (**D**) The mRNA expression of TNF-α and IL-6 in the intestine tissue of the WT and GSDME^−/−^ mice treated with or without cisplatin (*n* = 5). (**E**) The representative hematoxylin and eosin-stained (H&E) assay of the intestine tissue derived from the WT and GSDME^−/−^ mice (*n* = 5, scale bar: 100 μm). (**F**) The loss of surviving crypts and villus in the intestine samples derived from the WT and GSDME^−/−^ mice (*n* = 5). (**G**) The immunofluorescent staining of F4/80 (green, pan-macrophage markers), CD86 (M1 marker, magenta) or CD206 (M2 marker, red), GSDME (orange), and DAPI (nuclei, blue) of the intestine samples derived from the WT and GSDME^−/−^ mice (*n* = 3, scale bar: 100 μm). The average percentage of M1 and M2 macrophages were examined (*n* = 3). White arrows: F4/80^+^/CD86^+^ cells, yellow arrows: F4/80^+^/CD206^+^ cells. (**H**) The ratio of the percentage of M2 macrophages to that of M1 macrophages in the intestine tissue of WT and GSDME^−/−^ mice (*n* = 3). *p* values were calculated using one-way ANOVA with Dunnett’s test and unpaired Student’s t test. * *p* < 0.05, ** *p* < 0.01, *** *p* < 0.001, **** *p* < 0.0001.

**Figure 3 biomedicines-12-02526-f003:**
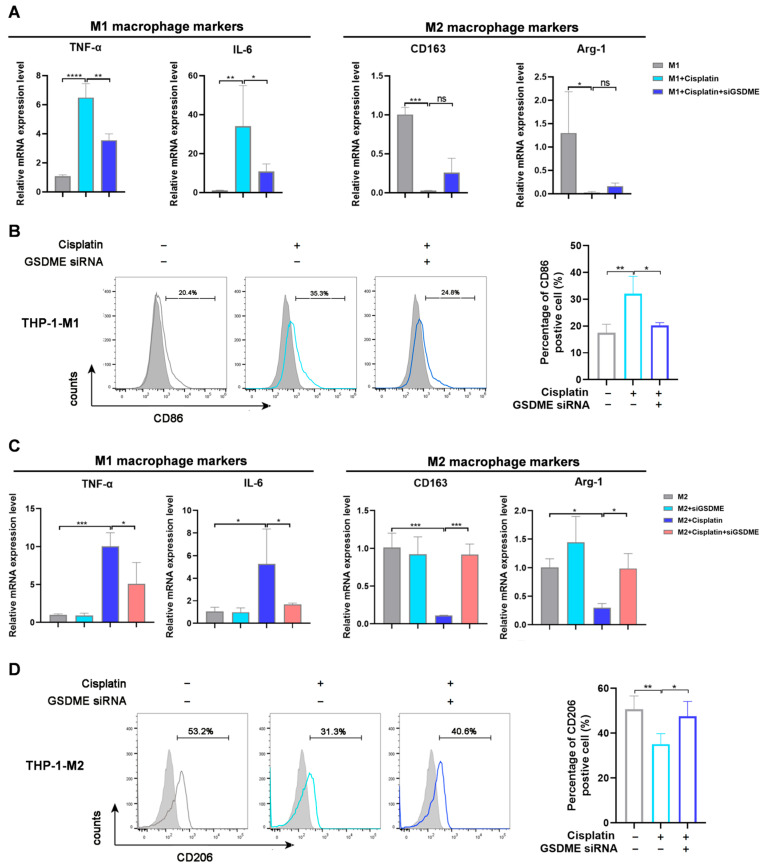
Cisplatin-induced M1 polarization was mediated by GSDME in vitro. (**A**,**C**) The mRNA expression levels of M1 (TNF-α and IL-6) and M2 (CD163 and Arg-1) markers in M1 (**A**) and M2 (**C**) macrophages upon cisplatin treatment. The experiments were conducted in triplicate (*n* = 3). (**B**,**D**) The variation in the percentage of M1-phenotype macrophages (**B**, CD86^+^) and M2-phenotype macrophages (**D**, CD206^+^) in different groups upon cisplatin treatment. Flow cytometry was used to analyze the CD86^+^ population in M1 macrophages and the CD206^+^ population in M2 macrophages (*n* = 3). *p* values were calculated using one-way ANOVA with Dunnett’s test. * *p* < 0.05, ** *p* < 0.01, *** *p* < 0.001, **** *p* < 0.0001, and ns (not significant).

**Figure 4 biomedicines-12-02526-f004:**
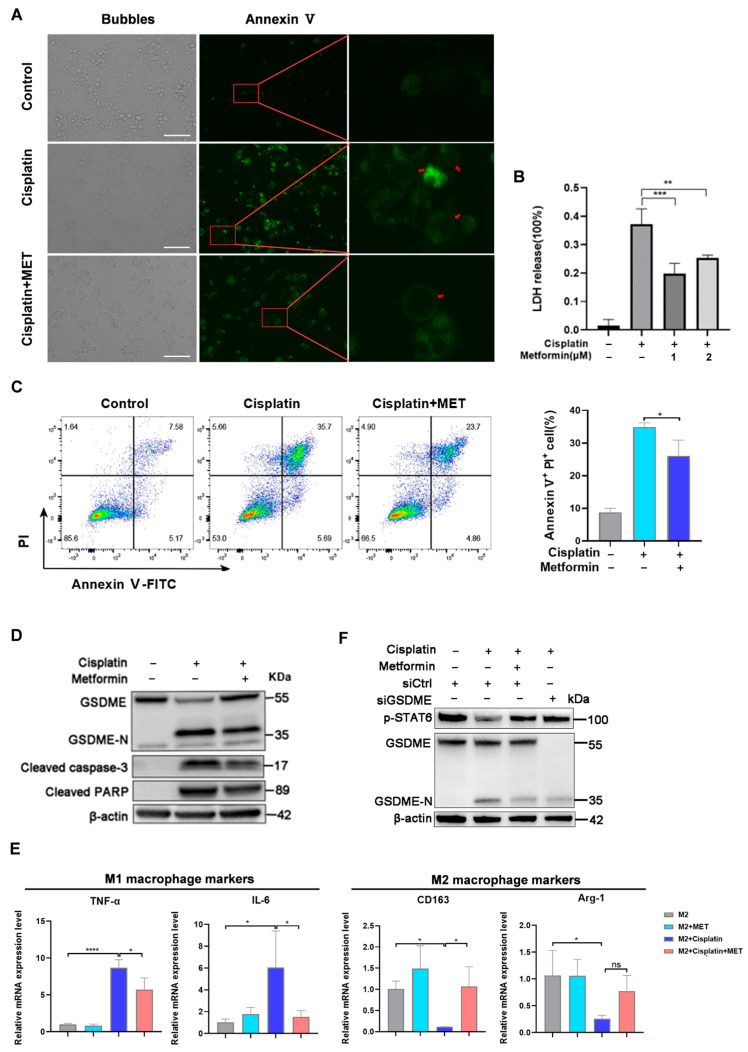
Metformin suppressed cisplatin-induced pyroptosis in M2 macrophages and maintained the phenotype of M2 macrophages. (**A**) Representative images of the immunofluorescent assay of the control M2 macrophages and cisplatin-treated M2 macrophages with or without the addition of metformin. Annexin Ⅴ-FITC was used to stain cell blebbing (red arrows). Scale bar, 50 μm. (**B**) The release of LDH from the control and cisplatin-treated M2 macrophages without or with metformin at different concentrations (*n* = 3). (**C**) Flow cytometry assay for Annexin V and PI in the control and cisplatin-treated M2 macrophages with or without metformin (*n* = 3). (**D**) Western blots of GSDME, activated caspase-3, and activated PARP expression in the control and cisplatin-treated M2 macrophages without or with metformin. (**E**) mRNA expression levels of M1 (TNF-α and IL-6) and M2 (CD163 and Arg-1) markers in the control M2 and cisplatin-treated M2 macrophages treated with or without metformin (*n* = 3). (**F**) Western blot analysis of GSDME and p-STAT6 expression in the control and cisplatin-treated M2 macrophages without or with metformin. *p* values were calculated by one-way ANOVA with Dunnett’s test. * *p* < 0.05, ** *p* < 0.01, *** *p* < 0.001, **** *p* < 0.0001, and ns (not significant).

**Figure 5 biomedicines-12-02526-f005:**
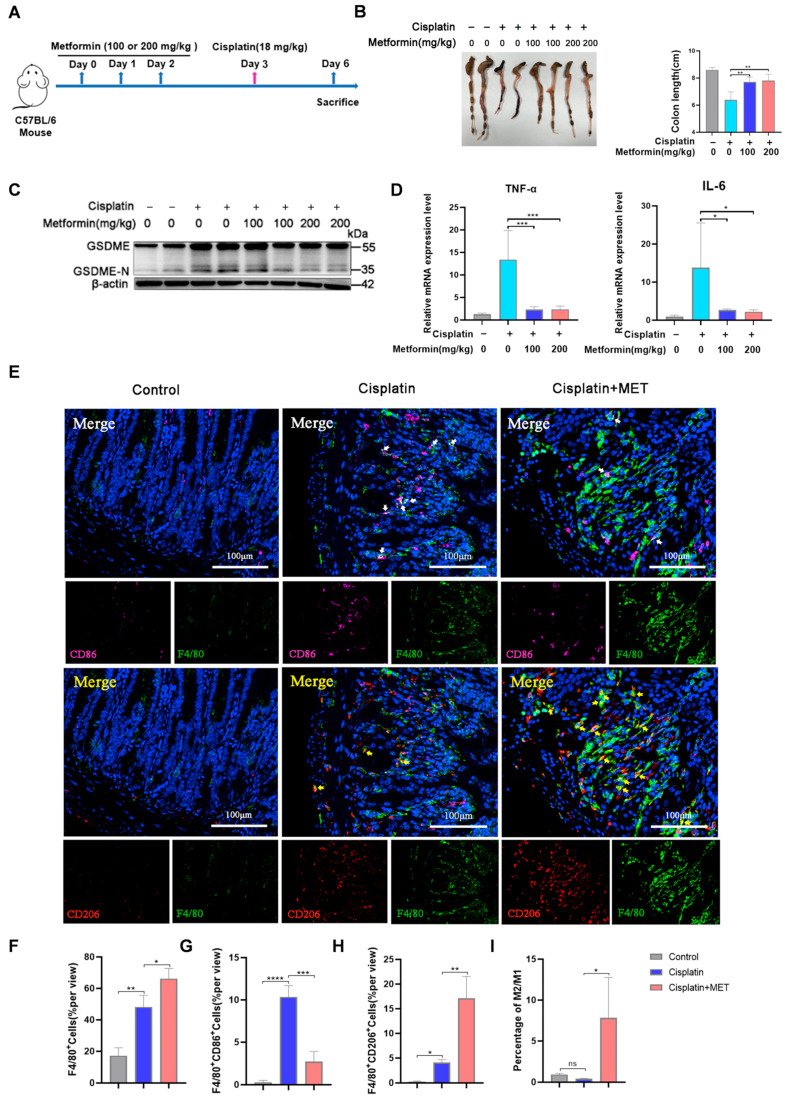
Metformin interrupted cisplatin-induced M1 polarization via GSDME in vivo. (**A**) Schematic of the treatment of cisplatin and metformin in mice models. (**B**) Representative images of the intestine tissue and the analysis of the colon lengths of different groups, as indicated (*n* = 6). (**C**) Western blot of GSDME expression of the intestine tissue derived from different groups, as indicated. (**D**) The mRNA expression of TNF-α and IL-6 in the intestine tissue of the control and cisplatin-treated mice with the addition of metformin at different concentrations (*n* = 5). (**E**) The immunofluorescence staining of F4/80 (green), CD86 (magenta) or CD206 (red), and DAPI (blue) on the intestine tissue sections of the control and cisplatin-treated mice with or without the injection of 200 mg/kg metformin for 3 days. Scale bar: 100 μm. White arrows: F4/80^+^/CD86^+^ cells, yellow arrows: F4/80^+^/CD206^+^ cells. (**F**–**I**) Statistical analysis of the percentage of F4/80^+^ (**F**), F4/80^+^/CD86^+^ (M1) (**G**), and F4/80^+^/CD206^+^ (M2) (**H**) cells and the ratio of M2 to M1 cells (**I**) in the intestine tissue of the control and cisplatin-treated mice with or without metformin (*n* = 3). *p* values were calculated by one-way ANOVA with Dunnett’s test and unpaired Student’s *t* test. * *p* < 0.05, ** *p* < 0.01, *** *p* < 0.001, **** *p* < 0.0001, and ns (not significant).

**Figure 6 biomedicines-12-02526-f006:**
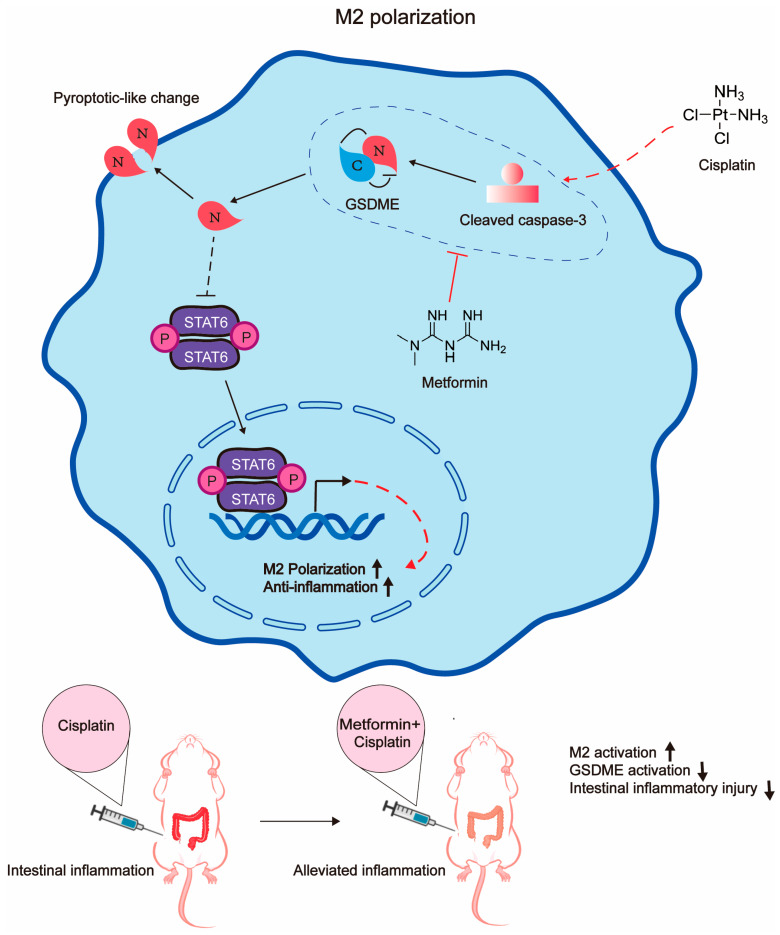
Schematic of the molecular mechanism of metformin-alleviated intestinal inflammation upon cisplatin treatment. Briefly, metformin inhibited the cisplatin-activated caspase3/GSDME pathway to induce M2 macrophage polarization, mitigating intestinal inflammation.

## Data Availability

The original contributions presented in the study are included in the article/Appendix A, further inquiries can be directed to the corresponding author.

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
