# Peer review of "Metformin Inhibited GSDME to Suppress M2 Macrophage Pyroptosis and Maintain M2 Phenotype to Mitigate Cisplatin-Induced Intestinal Inflammation"

_biomedicines, 2024, doi:10.3390/biomedicines12112526_

Round 1

Reviewer 1 Report

Comments and Suggestions for Authors

This is a well-written manuscript detailing the favourable effect of metformin in cisplatin toxicity.   1-The methodology part should be written in more details e g. Density of cells used in each experiment should be mentioned. Are the primers in PCR were pre-designed by the authors? How the authors checked their specificity. How did the authors check the specificity of products?

2-Why he authors chose gesdermin E but not gesdermin D?

Reviewer 2 Report

Comments and Suggestions for Authors

Thank you for the opportunity to review the manuscript “Metformin inhibited GSDME to suppress M2 macrophage pyroptosis and maintain M2 phenotype to mitigate cisplatin-induced intestinal inflammation” submitted to Biomedicines. The manuscripts focus on the molecular mechanism of metformin-mitigated intestinal damage induced by cisplatin. Overall, although this manuscript appears to be of good quality and provides valuable insights, some minor issues need to be addressed.

- Consider providing a diagram illustrating the mechanism by which metformin inhibits GSDME and suppresses M2 macrophage pyroptosis, maintaining the M2 phenotype to mitigate cisplatin-induced intestinal inflammation.

- What are the limitations of this study?

- It would be beneficial to discuss these results in relation to previous clinical findings.

Reviewer 3 Report

Comments and Suggestions for Authors

In this article, the authors first demonstrate that cisplatin can enhance the M1 phenotype. Then, using GSDME knockout mice, they show that GSDME knockout confers resistance to cisplatin-induced intestinal damage and increases M2 phenotype macrophages. Through animal experiments, they further prove that metformin alleviates cisplatin-induced intestinal inflammation by reducing M2 pyroptosis and enhancing the M2 phenotype via GSDME inhibition. This study is well-supported by data, has reliable conclusions, and shows strong innovation. It holds potential to provide valuable guidance for clinical treatments of intestinal injury diseases. 
